# Intervention Fatigue is the Primary Cause of Strong Secondary Waves in the COVID-19 Pandemic

**DOI:** 10.3390/ijerph17249592

**Published:** 2020-12-21

**Authors:** Kristoffer Rypdal, Filippo Maria Bianchi, Martin Rypdal

**Affiliations:** Department of Mathematics and Statistics, UiT the Arctic University of Norway, 9019 Tromsø, Norway; kristoffer.rypdal@uit.no (K.R.); filippo.m.bianchi@uit.no (F.M.B.)

**Keywords:** COVID-19, epidemic curve, second wave, intervention fatigue, reproduction number, SIR model, social response model

## Abstract

As of November 2020, the number of COVID-19 cases was increasing rapidly in many countries. In Europe, the virus spread slowed considerably in the late spring due to strict lockdown, but a second wave of the pandemic grew throughout the fall. In this study, we first reconstruct the time evolution of the effective reproduction numbers R(t) for each country by integrating the equations of the classic Susceptible-Infectious-Recovered (SIR) model. We cluster countries based on the estimated R(t) through a suitable time series dissimilarity. The clustering result suggests that simple dynamical mechanisms determine how countries respond to changes in COVID-19 case counts. Inspired by these results, we extend the simple SIR model for disease spread to include a social response to explain the number X(t) of new confirmed daily cases. In particular, we characterize the social response with a first-order model that depends on three parameters $\nu_{1},\;\nu_{2},\;\nu_{3}$. The parameter $\nu_{1}$ describes the effect of relaxed intervention when the incidence rate is low; $\nu_{2}$ models the impact of interventions when incidence rate is high; $\nu_{3}$ represents the fatigue, i.e., the weakening of interventions as time passes. The proposed model reproduces typical evolving patterns of COVID-19 epidemic waves observed in many countries. Estimating the parameters $\nu_{1},\;\nu_{2},\;\nu_{3}$ and initial conditions, such as $R_{0}$, for different countries helps to identify important dynamics in their social responses. One conclusion is that the leading cause of the strong second wave in Europe in the fall of 2020 was not the relaxation of interventions during the summer, but rather the failure to enforce interventions in the fall.

## 1. Introduction

The tendency of epidemics to return in repeated waves has been known since the 1918 Spanish flu [1], and the recent COVID-19 pandemic is no exception. In November 2020, the history of reported daily cases, or incidence rate, of COVID-19 varies considerably across the world’s regions. The broad picture is as follows [2]: The outbreak in China was practically over five weeks after a lockdown was imposed on January 23. Europe took over as the epicenter of the pandemic in late February. After lockdowns in most European countries in early March, followed by a gradual relaxation of these interventions, the first wave was over in the late spring. Incidence rates remained very low during the summer until they started to increase slowly in August. In late October, incidence rates higher than in March were common in Europe and grew exponentially with a week’s doubling time. The United States developed its first wave delayed by a week or two compared to Europe and a second and stronger wave throughout the summer. In November, the country was dealing with a third and even stronger wave. Many countries in South America, Africa, and South-East Asia were in the middle of (or had just finished) the first wave. On the other hand, a few countries such as New Zealand, Australia, Japan, and South Korea, had finished the second wave and managed to prevent it from becoming much stronger than the first one. Although there are a plethora of different wave patterns among the world’s countries, one could hope that these patterns fall into a limited number of identifiable groups.

In this paper, we do not claim that there is any epidemiological rule that states a pandemic evolving without social intervention must come in increasingly severe waves. On the contrary, the simple compartmental models devise the evolution of one single wave that finally declines due to herd immunity. We shall adopt the simplest of all such models here, the Susceptible-Infectious-Recovered (SIR) model, to describe the evolution of the epidemic state variables. However, in the SIR model, the effective reproduction number R, the average number of new infections caused by one infected individual, is proportional to the fraction of susceptible individuals *S* in the population. If initially $R=R_{0}>1$, the daily number of new infections (incidence rate) will increase with time *t* until *S* has been reduced to the point where R goes below 1, and then decay to zero as t→∞.

In November, herd immunity was not an essential mechanism in the COVID-19 pandemic because the fraction of susceptible individuals was still close to 1 in most populations. Consequently, the time variation of the reproduction number was predominantly caused by changes in social behavior. Changes in virus contagiousness could also play a rôle, but we have not taken virus mutations into account in this paper. Thus, by adopting the approximation S=1 in the definition of the reproduction number, the SIR model reduces to a set of two first-order ordinary differential equations for the cumulative number of infected cases J(t) and the instantaneous number of infectious individuals I(t). These are the “state variables” of the epidemic, which are driven by the reproduction number R(t). The evolution of the epidemic state [J(t),I(t)] can be computed as a solution to these equations if R(t) is known.

This paper’s philosophy is to make the simplifying assumption that R(t) responds to the epidemic state. More precisely, that the rate of change of R(t) is a function of the rate of change of J(t) depending on a set of parameters with distinct and straightforward interpretations that characterize the response. This mathematical relationship turns the SIR model into a closed model for the epidemic evolution, which depends on three parameters: (i) the relaxation rate when incidence rate is low, (ii) the intervention rate when incidence is high, and (iii) a fatigue rate that gradually weakens the effect of interventions over time. These parameters can be fitted to the incidence rate time series $X\left(t\right)=d_{t}J\left(t\right)$ reported by different countries. The analysis of the fitted values of these parameters, allows identifying groups of countries with similar evolution of the epidemics and help to understand the most effective mechanisms controlling the epidemic’s spread. One of our findings is stated in the paper’s title; intervention fatigue is the primary mechanism that gives rise to the strong secondary waves emerging in many countries.

The paper is organized as follows. Section 2.1 presents a method for reconstructing the R(t)-profile from the observed time series for the daily incidence rate using a simple inversion of the SIR model. The method’s effectiveness is illustrated in Section 2.2 by application to selected representative countries. In Section 2.3 we compute a dissimilarity measure between the reconstructed R(t)-profiles for each country in the world. Based on such a dissimilarity, we generate a dendrogram that hierarchically partitions countries according to their evolutionary paths of the epidemic. Finally, in Section 2.4 we construct a self-consistent, closed model for the simultaneous evolution of J(t) and R(t) and describe how this model can be fitted to the observed data for $d_{t}J$ for individual countries.

In Section 3.1 we synthesize the reconstructed R-curves for the majority of the world’s countries and use the dendrogram to group them into seven clusters, which are also shown on a World map. The features characterizing each cluster are analyzed and discussed. Section 3.2 illustrates the scenarios of the epidemic evolution that can be derived by solving the equations of the proposed closed model for different sets of parameters. The proposed model’s effectiveness is empirically validated in Section 3.3, where the model parameters are numerically fit to the incidence data for some selected countries exhibiting different characteristic patterns of epidemic evolution.

The possible implications of these results for COVID-19 strategic preparedness and response plans are discussed in Section 4.

## 2. Methods

### 2.1. Estimating the Reproduction Number from Incidence Rate Data

Let *S* be the fraction of susceptible individuals in a population, *I* the fraction of infectious, and *R* the fraction of individuals “removed" from the susceptible population (e.g., recovered, isolated, or deceased individuals). A simple model describing the evolution of these variables is the classical SIR-model [3],
(1)$$\begin{array}{ccc} \frac{dS}{dt} & = & -\beta IS, \end{array}$$
(2)$$\begin{array}{ccc} \frac{dI}{dt} & = & \beta IS-\alpha I, \end{array}$$
(3)$$\begin{array}{ccc} \frac{dR}{dt} & = & \alpha I, \end{array}$$
where α is the rate by which the infected are isolated from the susceptible population. Another interpretation of α is that $\alpha^{-1}$ is the average duration of the period an individual is infectious, which essentially depends only on the properties of the pathogen. As long as these do not change significantly, α will remain constant in time. In this paper we use α=1/(8days) but our results are not sensitive to this choice. The coefficient β, on the other hand, is the rate by which the infection is being transmitted. It evolves in time as societal interventions change. It is also influenced by behavioral changes in the susceptible population, such as eliminating superspreaders. The effective reproduction number is defined as
(4)$$R\left(t\right)\equiv \frac{\beta (t)}{\alpha }$$
and can be interpreted as the average number of new infections caused by an infected individual over the infectious period $\alpha^{-1}$.

The coupled system given by Equations (Equation 1) and (Equation 2), with initial conditions $S_{0}$ and $I_{0}$, constitutes a closed nonlinear initial value problem. Equation (Equation 3) is not a part of this system since it is trivially integrated to yield the removed population R(t) once I(t) is known.

The method developed in this paper is valid for an infectious disease with a new pathogen which is transmitted by contact between infectious and susceptible individuals. This implies that there is practically no immunity in the population from the start of the epidemic and we shall assume that this herd immunity is low throughout the period for which we estimate the reproduction number. In other words, we shall assume that S≈1, and hence that the cumulative fraction J=1−S of infected individuals is always much less than unity (J≪1). By introducing S=1−J in Equations (Equation 1) and (Equation 2), and by neglecting the term βIJ compared to the term βI in Equation (Equation 1), these two equations reduce to a linear model for *J* and *I*,
(5)$$\begin{array}{ccc} \frac{dJ}{dt} & = & \alpha R(t)I \end{array}$$
(6)$$\begin{array}{ccc} \frac{dI}{dt} & = & \alpha [R(t)-1]I\;. \end{array}$$

Please note that $\gamma_{I}\left(t\right)=I^{-1}dI/dt=\alpha \left[R\left(t\right)-1\right]$ is the relative growth rate for the instantaneous number of infectious individuals I(t), which is positive when R(t)>1 and negative when R(t)<1. By integrating Equation (Equation 6) and by inserting the result on the right hand side of Equation (Equation 5), we obtain that the daily number of new infections dJ/dt is determined by the initial $I_{0}$ and the history of R(t) on the interval (0,t);
(7)$$\frac{dJ}{dt}=\alpha I_{0}R\left(t\right)\exp\left(\int_{0}^{t}\alpha \left[R\left(t^{\prime }\right)-1\right)]dt^{\prime }\right).$$

What we are interested in here, however, is the inverse relationship; suppose the evolution of dJ/dt is known, how do we find the evolution of the reproduction number R(t)?

By using Equation (Equation 5) to replace αRI by $d_{t}J$ in Equation (Equation 6), the latter can be integrated to yield,
(8)$$I\left(t\right)=I_{0}e^{-\alpha t}+\int_{0}^{t}e^{-\alpha (t-t^{\prime })}d_{t^{\prime }}J\;dt^{\prime },$$
which allows us to compute R(t) from Equation (Equation 5);
(9)$$R\left(t\right)=\frac{d_{t}J}{\alpha I}=\frac{1}{\alpha }\frac{d_{t}J}{I_{0}e^{-\alpha t}+\int_{0}^{t}e^{\alpha (t^{\prime }-t)}d_{t^{\prime }}J\;dt^{\prime }}.$$

Provided a time series for J(t) is available, we can approximate $d_{t}J$ as a finite difference and the integral in Equation (Equation 9) as a discrete sum. This sum gives us a fast and direct algorithm to estimate R(t).

### 2.2. R(t)-Reconstructions for Individual Countries

Since we do not have actual measurements of the cumulative number of infected, to estimate the R(t)-curves for each country using Equation (Equation 9) we rely on the number of confirmed cases as a proxy for $d_{t}J\left(t\right)$. Specifically, we assume that the incidence rate $d_{t}J\left(t\right)$ is proportional to the daily number of confirmed cases X(t).

The time series X(t) of new daily cases reported for each country are taken from Our World in Data (https://ourworldindata.org/coronavirus-source-data). Figure 1 shows three examples of R(t) estimated using Equation (Equation 9) from the new daily cases X(t) reported by Sweden, Italy, and Argentina.

The initial values assumed by R(t) are affected by different choices of $I_{0}$. The transient effect given by the initial conditions quickly vanishes as *t* increases, and R(t) converges to a stable solution since the first term in the denominator of Equation (Equation 9) goes exponentially to zero. To compute the results, we generated several initial conditions for $I_{0}$ in a reasonable range, and we discarded the transient phase, depicted as a gray area in Figure 1. In these plots we have taken into consideration the delay between the date of infection and the reported positive tests, which may amount to approximately one week. Thus, the actual R(t) curves should be shifted towards the left by approximately this amount.

The three countries in Figure 1 are characterized by a different evolution of the epidemics. Sweden had a long first wave that peaked in the middle of June and the second wave started when the first one was not completely over. This is reflected in the estimated R(t), which stays for a long time interval above 1. Italy had a first wave stronger than other countries, that was brought down completely due to the lockdown. The estimated R(t) starts from very high values and quickly goes below 1 by the beginning of April. Finally, the number of new cases in Argentina kept growing very slowly, but consistently, until the middle of October and there are no two distinct waves as in many other countries. Consequently, the R(t) is characterized by values that are slightly above 1 until the beginning of November.

### 2.3. Cluster Analysis of R(t)-Curves

Rather than presenting reconstructions of R(t) case by case for all of the world’s countries, it would bring more insight to combine them in groups of R(t)-curves according to some common features and then analyze the characteristics of each group. Therefore, we follow such an indirect approach where we first cluster the R(t) curves of different countries and then analyze the clustering partition and the representatives of each cluster. The cornerstone of each clustering algorithm is the computation of a dissimilarity measure between the data samples. Since we are dealing with with sequential data, we leverage on a dissimilarity measure $\delta_{i,j}=d\left(x_{i},x_{j}\right)$ that yields a real number $\delta_{i,j}$ proportional to the discrepancy between the time series $x_{i}$ and $x_{j}$.

A large variety of time series dissimilarity measures have been proposed in the literature, including those based on statistical methods [4], signal processing [5], kernel methods [6], and reservoir computing [7]. In this paper, we adopt the Dynamic Time Warping (DTW) distance [8], which is an efficient and well-known algorithm that computes the dissimilarity between two sequences as the cost required to obtain an optimal match between them. The cost is computed as the sum of absolute differences between a set of indices in the two time series. DTW allows similar shapes to match, even if they are out of phase or, in general, not perfectly synchronized along the time axis.

From the dissimilarity $\delta_{i,j}$ between countries *i* and *j*, it is possible to compute a clustering partition, where similar R-time-series are assigned to the same cluster. Several approaches can be used to generate the clusters [9]. We opted for a hierarchical clustering method [10], which gradually joins data samples together by increasing the maximum radius of the clusters’ $\delta_{\max}$. One of the main advantages of hierarchical clustering is the possibility of generating a dendrogram, which allows visually exploring the structure of the clustering partition at different resolution levels.

### 2.4. A Closed Model for Model for the Epidemic Evolution

The SIR-model does not constitute a closed model for the evolution of J(t), I(t), and R(t). Equations (Equation 5) and (Equation 6) describe the dynamics of the epidemic state variables J(t) and I(t) when the evolution of the social state represented by R(t) is given. Equation (Equation 9) is nothing but an inverse of this relationship and should not be interpreted as a social response of R(t) to changes in the epidemic state variables. A closed model can only be obtained by adding an equation describing such a response. While a simple dynamical model cannot reflect the whole complexity of the social response, it may still provide some useful insight.

We shall represent this response by assuming that the rate of change $d_{t}R\left(t\right)$ is a function of the incidence rate $X\left(t\right)=d_{t}J\left(t\right)$, and that this function is positive when X(t) is below a threshold $X^{*}$ and negative when it is above that threshold. When the incidence rate is low, society responds by relaxing restrictions, and the reproduction number increases. When the incidence rate exceeds the threshold $X^{*}$, restrictions are introduced that make $d_{t}R\left(t\right)$ to change sign from positive to negative.

In the following, it is convenient to introduce a dimensionless time variable $t^{\prime }=\alpha t$, which allows us to formulate the differential equations as functions of the mean infectious time $\alpha^{-1}$ (which becomes the new time unit) rather than days. Accordingly, Equation (Equation 5) can be written as
(10)$$X\left(t^{\prime }\right)\equiv d_{t^{\prime }}J=RI,$$
and we have a closed model for $I(t^{\prime })$ and $R(t^{\prime })$ in the form of the dynamical system,
(11)$$\begin{array}{ccc} \frac{dR}{dt^{\prime }} & = & f(X)=f(RI), \end{array}$$
(12)$$\begin{array}{ccc} \frac{dI}{dt^{\prime }} & = & (R-1)I, \end{array}$$
where f(X) is assumed to be a differentiable function which is decreasing in a neighborhood of $X^{*}$ and with $f(X^{*})=0$. The system has a fixed point in R=1 and $I=X^{*}$. In this state, the number of infected stays constant at the threshold value.

By linearization of f(X) around the fixed point R=1 and $I=X^{*}$, and by introducing the rate constant $\nu =-\left(1/2\right)X^{*}f^{\prime }\left(X^{*}\right)>0$, the system reduces to
(13)$$\begin{array}{ccc} \frac{d\Delta R}{dt^{\prime }} & = & -2\nu (\Delta R+\Delta \tilde{I}+\Delta R\Delta \tilde{I}) \end{array}$$
(14)$$\begin{array}{ccc} \frac{d\Delta \tilde{I}}{dt^{\prime }} & = & \Delta R(1+\Delta \tilde{I}), \end{array}$$
where we have introduced ΔR=1−R and the normalized number of infected $\tilde{I}=I/X^{*}=1+\Delta \tilde{I}$. This nonlinear dynamical system has a stable fixed point in $\left(\Delta R,\Delta \tilde{I}\right)=\left(0,0\right)$.

#### 2.4.1. The Damped Harmonic Oscillator Model

In this section, we demonstrate that if *X* is close to the threshold value $X^{*}$ and R is close to 1, the linearization of Equations (Equation 11) and (Equation 12) leads to the equation for a damped harmonic oscillator. The purpose is to show analytically under what circumstances a damped oscillation is a natural time-asymptotic state of the epidemic. In Section 2.4.2 we argue that the model needs to be generalized to yield realistic descriptions of epidemic curves in most countries and, hence, the present section may be skipped without losing anything essential.

In the vicinity of the stable state $\left(\Delta R,\Delta \tilde{I}\right)=\left(0,0\right)$, linearization yields the damped, harmonic oscillator equation,
(15)$$\frac{d^{2}\Delta \tilde{I}}{dt^{\prime 2}}+2\nu \frac{d\Delta \tilde{I}}{dt^{\prime }}+\left(\omega^{2}+\nu^{2}\right)\Delta \tilde{I}=0,$$
where $\omega^{2}=2\nu -\nu^{2}$. For ν<2, the general solution is the damped oscillator
(16)$$\Delta \tilde{I}\left(t^{\prime }\right)=A\;e^{-\mu t^{\prime }}\cos\left(\omega t^{\prime }+\varphi \right),$$
where *A* and φ are integration constants and μ≡ν, and for ν≥2 the non-oscillatory strongly damped solution which for large $t^{\prime }$ goes as
(17)$$\Delta \tilde{I}\left(t^{\prime }\right)=Be^{-\mu^{\left(-\right)}t^{\prime }}+Ce^{-\mu^{\left(+\right)}t^{\prime }},$$
where *B* and *C* are constants of integration and $\mu^{\left(\pm \right)}\equiv \nu \left(1\pm \sqrt{1-2/\nu }\right)$. From Equation (14), we have
(18)$$\Delta R=\frac{d\Delta \tilde{I}}{dt^{\prime }},$$
and from Equation (Equation 10),
(19)$$\Delta X\left(t^{\prime }\right)=\Delta \left(\tilde{I}\left(t^{\prime }\right)R\left(t^{\prime }\right)\right)\approx \Delta R\left(t^{\prime }\right)+\Delta \tilde{I}\left(t^{\prime }\right)=\frac{d\Delta \tilde{I}}{dt^{\prime }}+\Delta \tilde{I},$$
which means that ΔI, ΔR, and ΔX experience the same damped oscillations with some phase shifts, or the same strongly damped solutions.

The frequency ω (for 0<ν<2) and the damping rate μ (for 0<ν<∞) are plotted against the parameter ν in Figure 1. The oscillation frequency and the damping rate are of comparable magnitude for ν<1, but the damping dominates in the interval 1<ν<2. For ν>2, the damping rate decreases towards 1 as ν increases.

The most rapid control of the epidemic is obtained when μ≈2, but we also have reasonably rapid control for larger ν. Slower damping takes place when ν<1, and slower the lower ν. Hence, what really should be avoided is ν much less than 1. The linearization of f(X) around $X^{*}$ in Equation (Equation 11) yields
(20)$$\frac{dR}{dt^{\prime }}=-\nu \Delta \tilde{X},$$
where $\tilde{X}=X/X^{*}$ is the incidence rate normalized to its threshold value. which shows that ν is a measure of how fast the rate of change in R responds to the deviation of $\tilde{X}$ from its threshold value ${\tilde{X}}^{*}=1$. If ν≪1, then $d_{t}R$ responds slowly to $\Delta \tilde{X}$, i.e., there is a slow social response to the rise or decay of the incidence. Equation (Equation 15) and Figure 2 then yields a damped oscillation with envelope that decays exponentially at a rate μ=ν. A characteristic duration of the epidemic is $\tau \equiv \nu^{-1}$ and the characteristic time scale of the oscillation is $T=\omega^{-1}=1/\sqrt{2\nu -\nu^{2}}$. Since the ratio between the two is $T/\tau =\sqrt{\nu /(2-\nu )}$, we observe that the oscillation scale *T* is longer than the decay time τ for all ν>1. Hence, this model suggests that oscillatory behavior and a long duration of the epidemic are features we expect to observe when the social response is slow (ν<1).

#### 2.4.2. A Nonlinear, Three-Parameter Oscillator Model

Although the linear oscillator model gives some insight into the mechanism that makes the epidemic return in repeated waves, there is an obvious lack of realism. One is to neglect the terms containing the product $\Delta R\Delta \tilde{I}$ in Equations (Equation 13) and (), since neither ΔR nor $\Delta \tilde{I}$ are, in general, small. There is also little reason to expect that the rate of change ν is the same below and above the social response threshold. Below the threshold, R increases because of the intervention’s termination and because the population relaxes. The more relaxed, the larger ν, so let us denote this parameter $\nu_{1}$ the “relaxation rate”. Above the threshold, R decreases because of the interventions aiming to strike the epidemic down. Stronger intervention translates into a larger “intervention rate” $\nu_{2}$. Even with these generalizations, the model will still give a damped, nonlinear oscillation and, hence, is unable to describe a situation where the second wave is stronger than the first. A generalization which may cover such a situation is to let the intervention rate decay with time, for instance, exponentially, such that we have an ultimate model for $\nu (t^{\prime })$ on the form
(21)$$\nu \left(t^{\prime }\right)=\nu_{1}\theta \left(-\Delta \tilde{X}\right)+\nu_{2}e^{-\nu_{3}t^{\prime }}\theta \left(\Delta \tilde{X}\right),$$
where θ(x) is the unit step function. The parameter $\nu_{3}$ can be thought of as a “fatigue rate”, i.e., the rate at which the strike-down rate is reduced because the population is becoming increasingly tired of interventions and restrictions.

Note also that the time dependence of the reproduction number in this model is independent of the response threshold $X^{*}$. This is because $X^{*}$ has been eliminated in Equations (Equation 13) and (Equation 14) through normalization of the variables. The un-normalized variables $I=X^{*}\tilde{I}$ and X(t) are, of course, proportional to $X^{*}$ and emphasizes the importance of a low tolerance threshold for social intervention.

#### 2.4.3. Fitting Model Parameters to the Observed Incidence Data

To validate the effectiveness of the proposed model in describing real data, we fit the three parameters $\nu_{1},\nu_{2},\nu_{3}$ with a numerical optimization routine that minimizes the discrepancy between the time series of reported new daily cases X(t) and those generated by the model. We constrained $\nu_{1}\in \left[0,\infty \right]$, while the other two parameters are unbounded. Besides the three model parameters, we also optimize with a grid search the following hyperparameters: the initial reproduction number $R_{0}$ searched in the interval [1.0,3.0] and the value $X^{*}=X\left(t^{*}\right)$ for each country with $t^{*}$ searched in the interval [15 January, 31 March]. As initial conditions for $\nu_{1},\nu_{2},\nu_{3}$ in the optimization routine, we used the values [0.1,0.1,0.1].

## 3. Results

### 3.1. Results of the Cluster Analysis of R(t)-Curves

The dendrogram to the left of Figure 3 depicts the result of the clustering procedure, based on the DTW dissimilarities between the R(t)-curves estimated according to Equation (Equation 9). In particular, the dendrogram illustrates how two leaves *i*, *j* (i.e., the R(t)-curves of countries *i* and *j*) are merged together as soon as the threshold $\delta_{\max}$ becomes larger than their DTW dissimilarity value $\delta_{i,j}$. There is no unique way of selecting an optimal $\delta_{\max}$, but it rather depends on what level of resolution of the clustering partition is amenable for a meaningful exploration of the structure underlying our data. In our case, we selected a $\delta_{\max}=150$ that gave rise to seven clusters, depicted in different colors in Figure 3. On the right hand side of Figure 3, we report R(t) averaged over all the countries in the same cluster. To facilitate the interpretation of the results, in Figure 4 we depict the same clustering partition obtained for $\delta_{\max}=150$ on the political world map.

The 1st cluster (light blue) contains countries mostly from Africa, South America and Middle East. The average R curve of the countries in the light blue cluster (top-right of Figure 3) shows that the reproduction number is always very low, but consistently above one. A possible explanation is that in those countries communities are more isolated and there are less travels and exchanges between them, making the infection to spread slower.

In 2nd cluster (green) the average R curve also stays always above one, but it starts from a higher value $R_{0}$. It is important to notice that this cluster includes large countries, such as India, Brazil, United States and Russia. In these countries, the time series of new cases X(t) have a particular profile since they are a combination from widely separated areas where the infection outbreak followed different courses. For instance, in the U.S., the waves in New York and California are almost in opposite phase.

The 3rd cluster (pink) contains countries where the first wave is very long and it took a considerable amount of time to bring the R curve below 1. A second wave is slowly emerging in the Northern autumn. An atypical member of this cluster is Sweden, which experienced a second wave in the summer that appeared almost as a continuation of the first wave, and then a strong third wave in the fall that is synchronous with the second wave for the rest of Western Europe (cluster 4).

The 4th cluster (brown) mostly contains Western European countries, characterized by a strong first wave that was brought down quickly and a second wave that begun in the fall. The average R curve is characterized by strong variability: it starts from a very high value and goes quickly below 1, to raise again quickly in the summer.

As with the 1st cluster, the 5th cluster (red) contains South American, African countries, and New Zealand. However, a key difference from 1st cluster is that in this case the R curve goes and remains below 1 during the Northern fall and autumn.

Finally, clusters 6 and 7 differ from the others by exhibiting initial R close to, or even lower than, one. For some countries in Cluster 6 this is an artifact of the averaging over all the countries in the cluster which includes some countries such as China, Australia, and South Korea, which started out with quite high R, but brought it down very rapidly through strong interventions [11]. The common characteristic feature for the cluster is an R(t) above 1 during the Northern summer, but a reduction in the fall, which is the opposite of what was observed in Western Europe and Canada. Cluster 7, on the other hand, contains most East-European countries, where the reproduction number was very low during the spring, but increased rapidly after the summer.

### 3.2. Exploring the Parameter Space of the Oscillator Model

The data for the incidence rate X(t) in the world’s countries show a wavy pattern consisting of one to three maxima during the first year of the pandemic evolution. However, the duration, relative strength, and separation between the waves vary substantially among countries and regions of the world. The total cumulative number of confirmed cases and deaths per million inhabitants can also vary by an order of magnitude or more among countries comparable to economic development, culture, and the healthcare system. Rypdal and Rypdal [12] demonstrated this for the first wave of the pandemic in a sample of 73 countries and discussed the significant differences in death toll between the two neighboring countries, Sweden and Norway. At the time of writing this paper, we are four months further into the pandemic. The picture has changed dramatically, with secondary and tertiary waves developing in many countries.

In Figure 5, we have summarized some of the conclusions drawn from numerical solutions of the model proposed in Section 2.4, obtained by varying the model parameters. In all simulations, we have chosen the time origin t=0 to be the first time the incidence rate X(0) crosses the threshold value $X^{*}$. Hence, $\tilde{X}\left(0\right)=1$ for all simulations. The incidence rate measured on the right-hand axis in the figures is measured in units of the threshold $X^{*}$.

#### 3.2.1. The Effect of the Initial Reproduction Number

In the first row of panels, Figure 5a–c, we consider the effect of changing the initial reproduction number $R_{0}$. From the reconstructed R(t)-curves, we observe that $R_{0}$ varies considerably among countries and regions. Low values just above $R_{0}=1$ are common in developing countries in South America, sub-Sahara Africa, and India. Several factors may contribute to this; lower mobility of people, a younger population, and a warmer climate. For these countries, we typically observe a slower rise and decay of the first wave, and the wave is generally weaker than in industrialized countries where $R_{0}$ varies in the range 2.0–2.5. In Figure 5a–c, we have changed $R_{0}$, keeping $\nu_{1}$, $\nu_{2}$, $\nu_{3}$ constant. By choosing $\nu_{1}=0$, and $\nu_{3}=0$ we consider countries that respond slowly to an incidence rate below the threshold and show little fatigue, which may be characteristic for developing countries for which Figure 5a may be relevant. In these panels, the choice $\nu_{2}=0.01$ is somewhat arbitrary but yields a rather stretched-out and low-amplitude first wave typical for those countries. For higher $R_{0}$, the first wave is higher in amplitude and shorter, like what we have seen in China. Here $\nu_{1}=\nu_{3}=0$ signify that the relaxation rate and fatigue have been sufficiently low to prevent R from increasing after it has stabilized below 1. The maximum incidence rate $\tilde{X}=X/X^{*}$ in panels (b) and (c) is high; in the range 15–40. In panel (i), where the parameters are the same as in (b) except for $\nu_{2}=0.1$ being ten times higher, shows $\tilde{X}\approx 3$, which, as we will see later, is representative for China.

#### 3.2.2. The Effect of the Relaxation Rate

In the second row, we vary the relaxation rate $\nu_{1}$ while keeping the strike-down rate fixed at $\nu_{2}=0.01$. The result is that as $\tilde{X}$ drops below the threshold after about 45 days, R(t) starts to rise and grow well beyond 1. How fast this happens, depends on $\nu_{1}$. In the phase when R>1, $\tilde{X}\left(t\right)$ will also start growing, and when it crosses the threshold $\tilde{X}=1$, the strike-down sets in again, and we enter a new cycle. With $\nu_{1}=0.01$ the first cycle takes almost 500 days, while it takes considerably less time in most countries, suggesting a higher $\nu_{1}$. In panel (e) we increase $\nu_{1}$ by a factor 5 and observe then two cycles within the first year, and in panel (f) another increment by a factor 10 almost eliminates the next waves. This faster relaxation to the equilibrium $R=\tilde{X}=1$ when the relaxation rate is high may appear counter-intuitive. After all, it leads to a rapid increase of R once $\tilde{X}$ has dropped below the threshold. However, the faster rise of R also leads to a faster rise of *X* beyond the threshold and to a faster strike-down of R back towards 1, i.e., to faster damping of the oscillation. This observation suggests that the strong second wave of the epidemic evolving in Europe in the fall of 2020 is not caused by the relaxation of social interventions during the summer but is caused by something else.

#### 3.2.3. The Effect of the Intervention Rate

A suspected candidate could be the intervention rate $\nu_{2}$, which is varied in the third row, panels (g)–(i). However, we observe that the main effect of increasing $\nu_{2}$ is to decrease the amplitude of the oscillation in $\tilde{X}$ in inverse proportion to $\nu_{2}$. In this row, we have kept $\nu_{1}=0$, resulting in relaxation to a time-asymptotic (t→∞) equilibrium $R_{\infty }<1$, ${\tilde{X}}_{\infty }=0$. This is in contrast to the second row ($\nu_{1}>0$), where this equilibrium is $R_{\infty }=1$, ${\tilde{X}}_{\infty }=1$. These two equilibria correspond to fundamentally different strategies to combat the epidemic. The one without the relaxation mechanism ($\nu_{1}=0$) corresponds to the strike-down strategy, where the goal is to eliminate the pathogen without obtaining herd immunity in the population. The one with $\nu_{1}>0$, allowing relaxation of interventions when the incidence rate dips below the threshold, will end up with a constant incidence rate at the threshold value and thus a linearly increasing cumulative number of infected until this growth is non-linearly saturated by herd immunity.

#### 3.2.4. The Effect of the Fatigue Rate

The effect of a non-zero fatigue rate is to bring the effective strike-down rate to zero as t→∞. The solution of the system Equations (Equation 13) and (Equation 14) as t→∞ is that $R\to 1+\nu_{3}$ and $\tilde{X}\approx \exp\left(\nu_{3}t\right)$. Of course, this blow-up is prevented by herd immunity, which will reduce the effective R to zero when most of the population has been infected. The effect of increasing immunity in the population is not included in Equation (Equation 11), and hence the model makes sense only as long as the majority of the population is still susceptible to the disease. Nevertheless, the last row in Figure 5 shows that increasing intervention fatigue represented by non-zero $\nu_{3}$ may increase the second and later waves’ amplitude and duration. For sufficiently large $\nu_{3}$ the second wave’s amplitude and duration can become greater than the first. The situations shown in panels (k) and (l) are observed in European countries and are caused by $\nu_{3}∼$0.1–0.2. One partial explanation of the second wave’s higher amplitude than the first, as observed in many countries, is a considerably higher testing rate. The testing rate, however, cannot explain the considerably longer duration of the second wave. This prolonged duration shows up both in the observed data and in this model, when the fatigue rate is increased.

### 3.3. Results from Fitting the Oscillator Model to Data for Selected Countries

In this section, we discuss the results obtained by fitting the proposed oscillator model fitted to the observed incidence data in the different World countries. Once again, we exploit cluster analysis to investigate the differences in values of the three fitted parameters $\nu_{1}$, $\nu_{2}$, and $\nu_{3}$ for each country.

This time, rather than computing the dissimilarity $\delta_{i,j}$ as the distance between the time series R(t) of country *i* and *j*, we let $\delta_{i,j}={\left|\log v_{i}-\log v_{j}\right|}^{2}$ where $v_{i}$ is a three-dimensional vector containing the parameters $\nu_{1}$, $\nu_{2}$, and $\nu_{3}$ fit on the country *i*. Such a cluster analysis allows visualizing the structure of the parameter space, where each data point represent a country. The log-transform in the computation of the dissimilarity allows better disentangling cluttered data points and to reduce the influence of outliers.

Figure 6 reports the partition obtained by thresholding the dendrogram of the hierarchical clustering at $\delta_{\max}=20$. We notice that some countries are not assigned to any cluster (depicted in white in the World map). The reason is the failure in converging of the optimization routine used to fit the model parameters, likely due to limited amount or irregularity of observed incidence data.

The partition contains six clusters and the largest ones are the azure and green clusters, containing mostly northern and southern countries, respectively. In Figure 7 we analyze in detail representative countries from each cluster, using for the graph borders the same color coding of Figure 6. Each graph in Figure 7 depicts the reported daily new cases (dashed red line), the daily new cases simulated by the oscillator model (solid red line), the R(t) curve estimated using Equation (Equation 9) (dashed blue line), and the R(t) curve simulated by the proposed close model (solid blue line). On the top of each graph, we report for each country the fitted values of $[\nu_{1}$, $\nu_{2}$, $\nu_{3}]$, the initial $R_{0}$, and the date $t^{*}$ that identifies $X^{*}=X\left(t^{*}\right)$. On the horizontal axis, 0 corresponds to $t^{*}$, the left vertical axis indicates the value of the reproduction number, the right vertical axis indicates the number of new daily cases. In the Appendix A, we report a visualization of $t^{*}$ in the different World countries, which can be interpreted as an estimate of the onset of the social responses.

The first row show results for two countries assigned to the first cluster, Brazil and India. Interestingly, Brazil and India were assigned to the same cluster (the one depicted in green in Figure 3 and Figure 4) also in the previous partition based on the time series dissimilarity of the reconstructed R(t). The initial reproduction number in these countries is low, $R_{0}=1.5$ for both countries, and the strike-down parameter is also low, $\nu_{2}\approx 0.01$, leading to a strong and long first wave, which is not yet completely over in November 2020. The fatigue rate of $\nu_{3}\approx 0.2$ also contributes to increasing the amplitude and the long-lasting downward slope of the first wave.

The second row show the typical pattern observed in Western countries, where a rather short first wave accompanied by a rapid drop in R(t) due to the almost universal lockdown in March 2020. Then, there is a rather slow relaxation of the interventions throughout the summer, finally leading to R stabilizing in the range 1.2–1.5. The inevitable result is the rise of a second wave, growing stronger and longer than the first, as shown in Figure 5k,l. In mid-November, interventions again had started to inhibit the growth, but they were weaker than in the spring, as reflected by the fatigue rates in the range $\nu_{3}∼$0.2–0.3. Indeed, the predictions of the oscillator model with the estimated rates is that the second wave will blow up in the spring of 2021 to levels where herd immunity will limit the growth. This is before vaccines are likely to play an important rôle, so a more probable scenario is that governments will reverse the fatigue trend and invalidate the model as a prediction for the future. Tendencies in this direction is observed in Europe at the time of writing. Differently from the partition described by the dendrogram in Figure 3 and by the map in Figure 4, most European countries are now assigned to the same cluster, along with US and Russia.

The 5th panel depicts the results for Ukraine, which has been selected as representative for the pink cluster. The most characterizing feature is that the reproduction number is not too high, but is never brought below one, resulting in a slow, but steady increment in the new daily cases. The extremely high $\nu_{2}$, combined with a correspondingly high $\nu_{3}$, reflects that the model in this case has problems describing the initial evolution of R(t). The effective intervention rate $\nu_{2}\exp\left(-\nu_{3}t\right)$ is negligible after a few days, so effectively X(t) grows exponentially without interventions with a growth determined by R(t)≈1.2.

The 6th panel depicts Turkey as representative of the orange cluster. The second wave for Turkey is not created by a finite fatigue rate, since $\nu_{3}=0$; it is created by a finite relaxation rate $\nu_{1}$. Importantly, this relaxation rate cannot create a second wave that is stronger than the first, it only gives rise to a damped oscillation that ends up in the equilibrium R=1, $X=X^{*}$. The model-fitted R(t) in Turkey grows slowly greater than 1, and a second wave in X(t) develops. This wave has an amplitude approximately the same as the first, but lasts longer (not shown in the figure), similar to what is shown in Figure 5k.

Argentina belongs to the purple cluster and the results are reported in the 7th panel. It shows a low initial reproduction number, but still larger than 1, which gives a slow growth of the epidemic, and a low intervention parameter that leaves R(t) at this level for a long time before it slowly forced below 1. As a result, the country ends up with a slow and strong first wave that has not reached its peak by November 2020.

Finally, the last panel describes the situation in China, which is a representative of the brown cluster. We notice that parameters estimated for China are comparable to those in Figure 5i. The peak incidence rate $X_{\max}\approx 3500$ for China is about three times the threshold incidence $X^{*}=1185$, similar to what is observed in Figure 5i. For China, the evolution of R(t) is initially rather similar to that of Turkey, and the shape of the X(t)-curve is also rather similar. However, in the Chinese case, the model-fitted R(t) converges to a fixed value $R_{\infty }<1$, and X(t) to 0 after a few months. This rise is the result of fundamental differences in the estimated model parameters: $\nu_{1}$ is zero for China but non-zero for Turkey. It may look confusing that R(t) for China estimated from the observed incidence rate (the dashed blue curve), deviates from the theoretical curve and is above 1 during long periods. This is because in China, after the first wave, the incidence rate was so low that it is impossible to estimate R(t) accurately. Indeed, the confidence interval in the estimate of R(t) is very large, but we have not bothered to report it because it would clutter up the figures.

## 4. Discussion and Conclusions

The geographic distribution of countries belonging to different clusters shown in the map in Figure 4, and the associated averaged R(t)-curves in Figure 3, may serve as a crude road map to the global evolution of the pandemic throughout the spring and fall of 2020. One striking feature is some geographic clustering, which is most pronounced in Western Europe (brown) and Eastern Europe (blue). A similar clustering is seen in the U.S. and Equatorial Latin America (green). In this paper, we have a focus on the strength, timing and duration of the second epidemic wave, and for this purpose the dendrogram helps us to identify those regions where there has been a pronounced second wave so far in the pandemic. These are those countries that belong to clusters exhibiting a period of R(t)<1 in between periods of R>1. From the R(t)-profiles in Figure 3 those countries with the most pronounced second wave are Cluster 4 (brown) and 7 (dark blue), Western and Eastern Europe, respectively. The rise of the second wave here is due to the persistently high values of R during the period July–November. What distinguishes the two clusters is the course of the first wave. In Western Europe there was a strong first wave associated with high R, and it affected strongly older age groups which resulted in high case fatality ratio (CFR). The second wave has affected all ages and so far the death numbers have been much lower than in the first. In Eastern Europe the first wave was very weak, but the second has been strong and with considerably higher CFR than in the countries further West.

The main result in this paper is that it demonstrates that the varying courses of the epidemic depicted via the seven characteristic R(t) curves shown in Figure 3 to some extent can be understood in terms of the interplay between three social responses to the epidemic activity; the relaxation of interventions when the activity is low, the intensification of interventions when activity becomes high, and the intervention fatigue which develops with time. Figure 5 and Figure 7 suggest that most country-specific epidemic curves can be qualitatively reproduced by a simple mathematical model involving these three responses. The value of this insight is that, despite the immense complexity and diversity of the dynamical response triggered by this new pathogen, there are some universal governing principles that will determine the final outcome in the years to come.

Our analysis suggests that a necessary condition for the development of a strong second wave is the absence of resolute response when it becomes clear to everybody that the reproduction number is rising well beyond 1. With no intervention fatigue ($\nu_{3}=0$), our model cannot produce a second wave more severe than the first, even with a very high relaxation rate $\nu_{1}$ resulting from summer holidays in the Northern hemisphere. This is actually quite obvious without modeling. The first wave started when there were no interventions and extensive winter tourism in Europe. We never returned to those favourable conditions for virus spread during the summer (i.e., R never returned to the high values of late February), so if Europe had imposed the same restriction in the fall as in the spring the second wave could not have been more severe than the first. The fact that it did grow stronger in terms of incidence rate, reflects the fact that interventions in the fall up to mid-November have been very weak, i.e., we have seen the effect of intervention fatigue.

It is of course true that the second wave would have been avoided all together if strong restrictions had been maintained throughout the summer, as in China. Then R would have remained less than 1, as shown in Figure 5a–c. However, this was not the case in most countries, as restrictions were lifted when incidence rates dropped low. The reproduction number rose above 1 during the Northern summer almost everywhere, but Figure 5d–e shows that our model does not predict a larger second wave if this happens early. This seems to be consistent with what was observed up to mid-November in Europe.

The model devised here could of course be run to make projections further ahead than one year from the onset of the epidemic, as done in Figure 5. It would show a blow-up of all solutions for which the fatigue parameter is non-zero, and would be unrealistic for several reasons. One is that the linearity approximation would break down as herd immunity will start to bring the effective reproduction number down. Another is that the intervention fatigue model most likely will fail when the epidemic activity becomes sufficiently high. We have already have seen signs in this direction in many European countries where partial lockdowns and mass testing again have succeeded in “bending the curve” to an extent that is not described by the model. Finally, mass-vaccination will hopefully become a real game-changer in the year to come.

## Figures and Tables

**Figure 1 ijerph-17-09592-f001:**
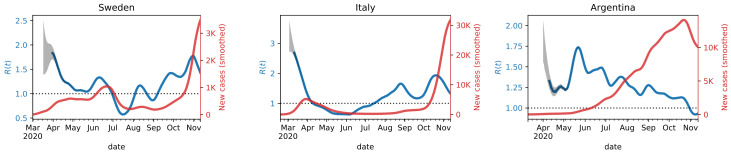
R(t) estimated with the proposed method from the time series of new daily cases reported in Sweden, Italy, and Argentina. Red curves are reported incidence rates, and the blue curves the reconstructed reproduction numbers.

**Figure 2 ijerph-17-09592-f002:**
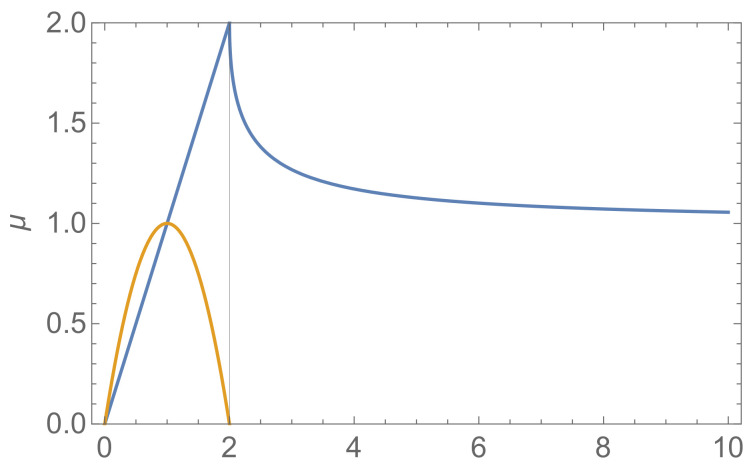
The yellow curve shows the frequency $\omega =\sqrt{2\nu -\nu^{2}}$, and the blue curve the damping rate μ=ν for ν<2 and $\mu =\nu (1-\sqrt{1-2/\nu }-1)$ for ν≥2.

**Figure 3 ijerph-17-09592-f003:**
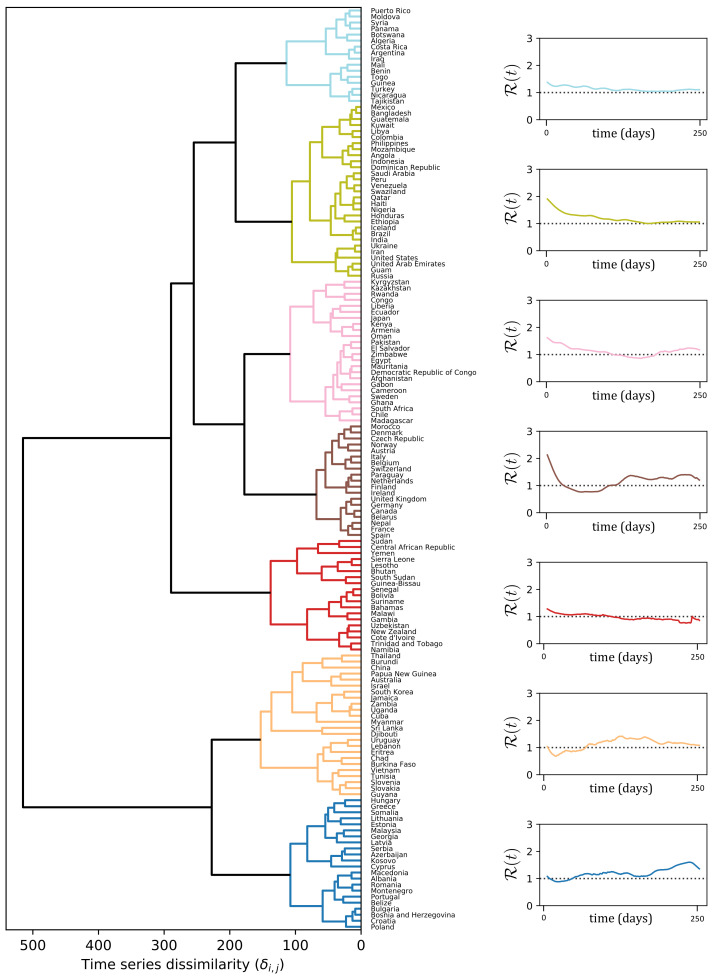
The left figure depicts the dendrogram obtained from the DTW dissimilarity between the R(t) time series. It is possible to obtain a certain number of clusters by putting a threshold at a specific dissimilarity value. In the example, we choose the threshold equal to 130. The right figures depict the average R(t) of countries in the same cluster.

**Figure 4 ijerph-17-09592-f004:**
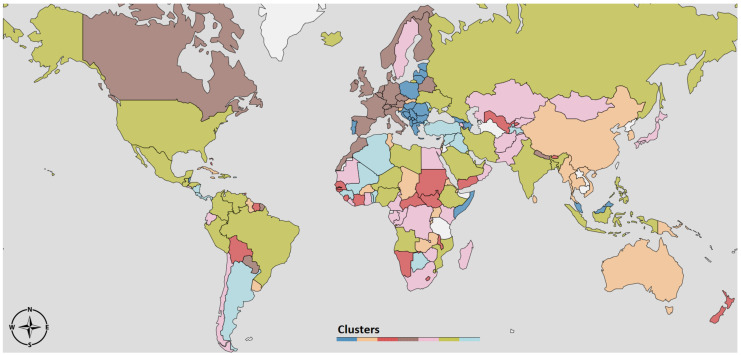
Visualization on the World map of the clusters obtained from the dissimilarity of the R(t) curves.

**Figure 5 ijerph-17-09592-f005:**
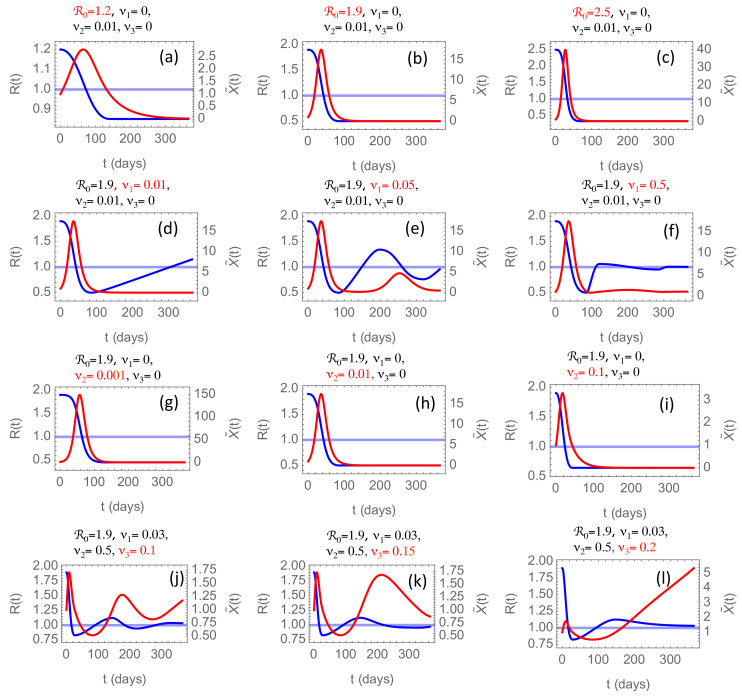
Blue curves show the evolution of R(t) and red curves $\tilde{X}\left(t\right)$ as solutions of Equations (Equation 13) and (Equation 14) with $\tilde{X}=R\tilde{I}$ and ν given by Equation (Equation 21), initial conditions $\tilde{X}\left(0\right)=1$, $R\left(0\right)=R_{0}$, and parameters $R_{0},\;\nu_{1},\;\nu_{2},\;\nu_{3}$ as indicated in the figures. A detailed description of each subfigure can be found in Section 3.2.1, Section 3.2.2, Section 3.2.3 and Section 3.2.4.

**Figure 6 ijerph-17-09592-f006:**
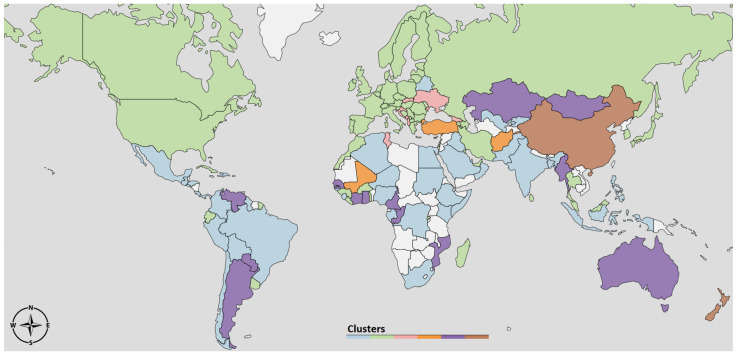
Visualization on a world map of clusters obtained from the dissimilarities of the vectors containing the fitted model parameters $\nu_{1}$, $\nu_{2}$, and $\nu_{3}$.

**Figure 7 ijerph-17-09592-f007:**
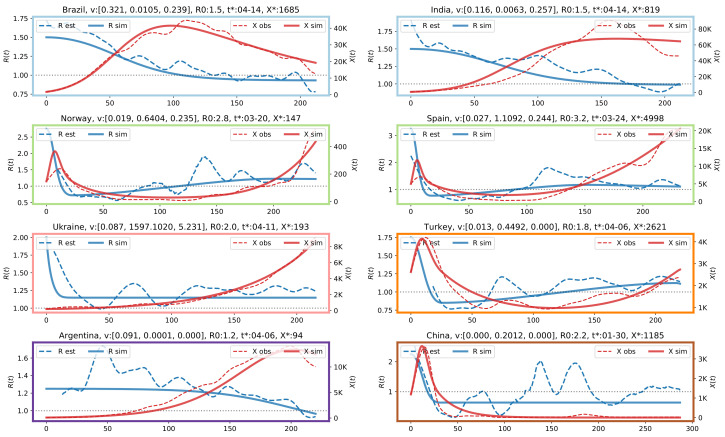
Comparison between observed daily cases X(t) (red dashed lines), X(t) simulated from model (solid red lines), R(t) estimated from the data using the inverted SIR model (dashed blue line), R(t) simulated from the model.

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
