# Peer review of "Intervention Fatigue is the Primary Cause of Strong Secondary Waves in the COVID-19 Pandemic"

_ijerph, 2020, doi:10.3390/ijerph17249592_

Round 1

Reviewer 1 Report

In the manuscript “Intervention fatigue is the primary cause of strong secondary waves in the COVID-19 pandemic”, the authors developed a new model system linking virus transmission with different response rates (i.e. intervention intensity). It was highlighted that intervention fatigue caused the intervention intensity to decrease, so that the second wave of COVID-19 in many countries are so high. The authors did sensitivity analysis for the key parameters, and matched their model with confirmed cases. I feel the value of this study is that the complex control measures against COVID-19 were simplified as three response rates based on the traditional SIR model, which well distinguished different control policies in all the countries (demonstrated by the cluster analysis and Figure 5&6). I have some suggestions listed below:

  1. Line 11, “X*” should be “X”。
  2. Line 13, the fatigue function is an exponential function, which drops faster in the beginning. I feel a linear function is more reasonable, and a S-shaped curve (logistic curve) is most likely to match the real situation. Under a social distance policy, there is usually a few people violate the regulation in the beginning. As time goes by, more people need to go out. Whereas there are always some people strictly follow the regulation all the time. Justification for the exponential function is needed.
  3. Line 36, the situation of COVID-19 in Japan is different from that in Australia, New Zealand, and South Korea. Japan has over 1000 new confirmed cases every day in past month, and has implemented travel and meeting restrictions for many places. Japan should be removed here.
  4. Line 49-50. Defining the proportion of susceptible individuals to be 100% simplifies the model. I feel it is acceptable, and I know it cannot be changed in this study. However, the virus has transmitted on the earth for one year, and the recovered population has grown in many countries. For example, near 5% of the whole population in the United State has been confirmed to be infected, and the actual infection rate should be 20% or even higher. For many countries in Europe, the cumulative cases have reached 3% of the population. As such, the current stage is not the beginning of virus transmission. More words are needed to justify such setting.
  5. Line 75, replace “Section ??” by “Section 2.2”. I guess this error was caused by WORD-PDF conversion. You may adjust line width, or add a space around “Section 2.2”, to overcome such error.
  6. It is a very good idea for doing a cluster analysis to group countries with similar virus control policies. Time series R(t) is a good index that represent control policies. Another index, the cumulative cases may also provide meaningful result.
  7. I found the R(t) for China (Figure 6, blue dashed line based on reported confirmed cases) is not correct. It shows two peaks at days from 130 to 180. China does not have such peaks for R(t). China implements super strict control measures and does not tolerate a single local infection. Given a few imported cases in Beijing, Dalian, Qingdao, Yili, Manzhouli, Shanghai, millions of virus detection were completed in a few days without any delay, so that local transmission was terminated at the early stage. I guess the authors did not distinguish imported cases and locally transmitted cases. The imported cases are much higher than local cases in China. Because the number of daily new cases is very low, the R(t) would be high if some imported cases was mixed. If the cumulative cases is used for cluster analysis, such error would be decreased. I am not suggesting authors to change R(t) in the cluster analysis, I just provide more options.
  8. Line 183, remove one “model for” in the section title.
  9. If the authors can estimate the values for v1, v2, v3, and X* for all countries, and carry out a hierarchical cluster analysis using the three variables (v1, v2 and v3), it would be very nice because all countries are grouped based on their control policy intensity using the authors’ model system. It is OK that content is not added into the manuscript, since Figure 5 and figure 6 have provided similar results. I like the two figures very much, and I expect to see more countries in Figure 6.

Author Response

A pdf file with the response to both reviewers is attached. 

Reviewer 2 Report

I have a controversial view on this paper.

From a technical standpoint, e.g., management of the SIR regression model, focus on the Covid fatigue, clarity of purposes ... etc, this seems to be a good paper, with a positive potential for publication.

Nonetheless, if I a move to the most important points that are treated I believe there is an underestimation of many factors that cannot be accepted.

Take the conclusions: "One conclusion is that the leading cause of the strong second wave in Europe in the fall of 2020 was not the relaxation of interventions during the summer, but rather the general fatigue to interventions developing in the fall."

There is plenty of literature that deny those statements.

First example: the vacations are a crucial moment for Covid spreading both in winter and in summer, unfortunately; I know for sure about recent papers that are going to be published with regard to the negative potential of the 2020 summer holidays. This is not a scientific paper, yet it well describes the situation for Italian island of Sardinia (almost no infection before summer, hundreds of cases at the end of summer):

Giuffrida, A. How Sardinia went from safe haven to Covid-19 hotspot. The Guardian 2020.

Not to mention the role played by winter vacations at the beginning of the pandemic in Europe, as witnessed by this paper:

Falk, M.T.; Hagsten, E. The unwanted free rider: Covid-19. Current Issues in Tourism 2020, 1–6, doi:10.1080/13683500.2020.1769575.

Not only, but the role of tourism and travels for the virus spread is a matter of fact, as well explained here:

Farzanegan, M.R.; Gholipour, H.F.; Feizi, M.; Nunkoo, R.; Andargoli, A.E. International Tourism and Outbreak of Coronavirus (COVID-19): A Cross-Country Analysis. Journal of Travel Research 2020, 0047287520931593, doi:10.1177/0047287520931593

Second example: it seems that one of the authors is Italian, then he should know very well that it is not true that none had predicted this second surge of the virus, at least In Italy, as witnessed in this paper published before the end of August 2020:

Mirri, S.; Delnevo, G.; Roccetti, M. Is a COVID-19 Second Wave Possible in Emilia-Romagna (Italy)? Forecasting a Future Outbreak with Particulate Pollution and Machine Learning. Computation 2020, 8, 74, doi:10.3390/computation8030074.

In conclusion, we should begin to admit that this second wave, that is hitting violently many European countries, was well expected by scientists, while those who are in charge of taking the decisions (politics) have been more influenced by the pressure of the economy to reopen.

At the end, I would like to see this paper revised taking into account, at least in an adequate discussion with precise references to the aforementioned papers, that some real facts have happened that have led us up to this dramatic point, even if their mathematical model is not fully able to account for all this factors together.

Revised this way the paper could have my approval.

Author Response

A pdf with the response to both reviewers is attached. 

Round 2

Reviewer 2 Report

I have no point in favor of this paper, as all the concerns I had raised in my first review have been systematically ignored or neglected, with no change in the paper.

In my first review, I had said that there was a potential in the paper, provided that the authors have (at least) recognized that fatigue cannot be the only predictor of the second spread of this pandemics. I had also gave them some points on which their discussion could be, at least, extended. Yet, they have rejected all this.

Since they are refusing to use both scientific argumentations and also common sense, I am extremely negative about this paper.

It cannot be published in this present form, as it represents a form of ideology (hidden by a mathematical model) and not true science.

This paper has to be rejected.

Author Response

We are very surprised about this second report.  

Reviewer:  “All the concerns I had raised in the first review has been systematically ignored or neglected and no change in the paper.” 

Rebuttal: Nothing in this reviewer’ report has been ignored. In fact, we recognized to be in agreement with the reviewer’s points and we reformulated our claims to clarify this. We have reformulated the abstract to clarify that we don't interpret intervention fatigue as pertaining to individual responses to government interventions, but rather the absence or delay of government intervention. We have also included two paragraphs in the Discussion (Section 4) to address the concerns of Reviewer 2. 

Reviewer: In my first review, I had said that there was a potential in the paper, provided that the authors have (at least) recognized that fatigue cannot be the only predictor of the second spread of this pandemics. I had also gave them some points on which their discussion could be, at least, extended. Yet, they have rejected all this. 

Rebuttal: We do not claim that “fatigue is the only predictor of the second spread of the pandemic.” We claim that it is the primary driver of a strong second wave. This is justified by our modeling and its interpretation in the paper, and explained thoroughly in our first response and enforced in the revision.  The “points on the discussion” given by the referee was that tourism contributed to the spread of the incidence in Europe throughout the summer. But that is not in contradiction with our model or our conclusions. The “relaxation rate” in our model is supposed to represent that effect, and the importance of that effect for the development of the second wave was studied in the model and given common-sense interpretations. 

Reviewer: Since they are refusing to use both scientific argumentations and also common sense, I am extremely negative about this paper. It cannot be published in this present form, as it represents a form of ideology (hidden by a mathematical model) and not true science. 

Rebuttal:  This kind of inflammatory allegation has no place in a scientific review process.